# Multi-trajectory analysis of changes in physical activity and body mass index in relation to retirement: Finnish Retirement and Aging study

**Roosa Lintuaho**[1,2]*, **Mikhail Saltychev**[2], **Jaana Pentti**[3,4,5], **Jussi Vahtera**[3,4], **Sari Stenholm**[3,4]

1 Social and Health Services, Kaarina, Finland, 2 Department of Physical and Rehabilitation Medicine, Turku University Hospital and University of Turku, Turku, Finland, 3 Department of Public Health, University of Turku and Turku University Hospital, Turku, Finland, 4 Centre for Population Health Research, University of Turku and Turku University Hospital, Turku, Finland, 5 Clinicum, Faculty of Medicine, University of Helsinki, Helsinki, Finland

* roemti@utu.fi

## Abstract

### Background

Physical activity and body mass index (BMI) have been reported to change around retirement. The objective was to examine the concurrent changes in physical activity and BMI around retirement, which have not been studied before. In addition, the associations of different demographic characteristics with these changes were examined.

### Methods

The prospective cohort study consisted of 3,351 participants in the ongoing Finnish Retirement and Ageing Study (FIREA). Repeated postal survey, including questions on physical activity and body weight and height, was conducted once a year up to five times before and after the retirement transition, the mean follow-up time being 3.6 years (SD 0.7). Group-based multi-trajectory modeling was used to identify several clusters with dissimilar concurrent changes in physical activity and BMI within the studied cohort.

### Results

Of the participants, 83% were women. The mean age at the last wave before retirement was 63.3 (SD 1.4) years. Four clusters with different trajectories of physical activity and BMI were identified. BMI remained stable around retirement transition in all four clusters, varying from normal weight to class II obesity. The association of BMI trajectories with physical activity levels were inverse, however, each activity trajectory showed a temporary increase during the retirement transition.

**Data Availability Statement:** The dataset supporting the conclusions of this article were obtained from the FIREA study. The FIREA dataset comprises health related participant data and their

use is therefore restricted under the regulations on professional secrecy (Act on the Openness of Government Activities, 612/1999) and on sensitive personal data (Personal Data Act, 523/1999, implementing the EU data protection directive 95/46/EC). Due to these legal restrictions, the data from this study cannot be stored in public repositories or otherwise made publicly available. However, data access may be permitted on a case by case basis upon request for bona fide researchers with an established scientific record and bona fide organisations. Data sharing outside the group is done in collaboration with FIREA group and requires a data-sharing agreement. Investigators can submit an expression of interest to FIREA research group at University of Turku, Finland (firea@utu.fi).

**Funding:** This study was supported by funding granted by the Academy of Finland (321409 and 329240 to JV, 286294, 319246, 294154, 332030 to SS), Finnish Ministry of Education and Culture (to SS); Juho Vainio Foundation (to SS), and Hospital District of Southwest Finland (to SS) The funder had no role in study design, data collection and analysis, decision to publish or preparation of the manuscript.

**Competing interests:** The authors have declared that no competing interests exist.

**Abbreviations:** BMI, Body Mass Index; FIREA, Finnish Retirement and Ageing Study; MET, Metabolic Equivalent of Task; ISCO, International Standard Classification of Occupation; GBTA, Group Based Multi-trajectory Analysis.

## Conclusions

Retirement seems to have more effect on physical activity than BMI, showing a temporary increase in physical activity at the time of retirement.

## Introduction

Retirement is a major event in late midlife altering daily routines since work no longer dominates the everyday schedule. Among other changes, retirement may affect physical activity and weight control due to the absence of occupational and commuting physical activity and altered leisure-time physical activity, eating pattern and diet [1,2].

A study from the United States has shown that both aging and retirement might associate with an increase in body mass index (BMI) [3]. This increase related to retirement has been found to be greater among manual workers compared to those retiring from sedentary jobs. Another study from the United States has found a 5% weight gain in women associated with retirement, but no respective change has been found in men [4]. Similarly, a Finnish study has reported divergent changes in women and men, so that weight increased in women retiring from physically heavy jobs and decreased in men retiring from sedentary jobs [5]. Moreover, a Dutch study has found that workers retiring from manual jobs gained more weight than those who stayed at work or those retiring from sedentary jobs. On the other hand, those retiring from sedentary jobs gained less weight than those remaining at work [6].

Retirement has also been shown to affect physical activity [1]. A French study has found an increase in self-reported leisure-time activity and sport-related activities after the retirement especially among women [7]. Similarly, a Finnish study has observed increase, albeit temporary, in self-reported leisure-time physical activity at a moderate-intensity level around retirement [8]. Using accelerometer measurements, another Finnish study has found that daily total physical activity decreased among women retiring from manual work, but not in women retiring from sedentary work [9]. Respectively, men retiring from sedentary work have shown increased physical activity, while such an effect has not been observed among men retiring from manual jobs.

Even though physical activity and BMI are strongly associated, no previous study has investigated the simultaneous changes in physical activity and BMI in relation to a retirement transition. Thus, the aim of this study was to examine the concurrent changes in physical activity and BMI around retirement using a multivariate trajectory analysis. The study focused on identifying groups that might demonstrate different trajectories of simultaneous changes in physical activity and BMI around retirement. Since gender, socioeconomic status and marital status have been found to be associated with changes in both physical activity [1,8] and BMI during retirement transition[3], the aim was also to examine whether gender, occupational status and marital status predict the probability of being classified into a particular cluster.

## Methods

### Study population

The Finnish Retirement and Aging Study (FIREA) is an ongoing longitudinal cohort study among older adults. The aim of the FIREA study is to follow ageing workers from working life until a full-time retirement and to determine how health behaviours and clinical risk factors are changing during that transition. The eligible population for the FIREA included all public sector employees whose individual retirement date was between 2014 and 2019 and who were

working in the year 2012 in one of 27 municipalities, nine cities or five hospital districts. Information on the estimated individual retirement date was obtained from the comprehensive register kept by the Finnish largest pension provider for public sector employees (Keva).

The participants were first contacted 18 months prior to their estimated retirement date by a questionnaire, which has been thereafter sent annually, at least four times. The actual retirement date was reported by the participants. Due to the eligibility criteria, large majority of the FIREA participants retired based on their age, and not due to a disease. The FIREA study have been described in detail earlier [10].

To be included in the current study, the participants had to be still working during the first questionnaire and to have provided information on physical activity and BMI at least two consecutive times, one right before and one right after the retirement. Of the FIREA cohort members, 6,679 (63% of the eligible sample of 10,629 employees) had responded at least once by the end of 2019. Of these respondents, 5,076 were still working at the time of first response. Of these, at least two consecutive responses, one before and another after the retirement, were available from 3,426 participants. The information on physical activity and BMI immediately before and after the retirement was available from 3,351 respondents. There were two possible survey waves before the retirement, waves -2 (approximately 18 months prior to estimated retirement date) and -1 (approximately 6 months prior to estimated retirement date), and three possible waves after the retirement, waves 1 (approximately 6 months after retirement), 2 (approximately 18 months after retirement) and 3 (approximately 2.5 years after retirement). (Fig 1). Each successive wave was one year apart from each other. On average, the respondents provided information on physical activity and body mass index at 3.6 (SD 0.7) of the possible study waves.

All the respondents have provided a written informed consent. The FIREA is following the Declaration of Helsinki and has been approved by the Ethics Committee of Hospital District of Southwest Finland.

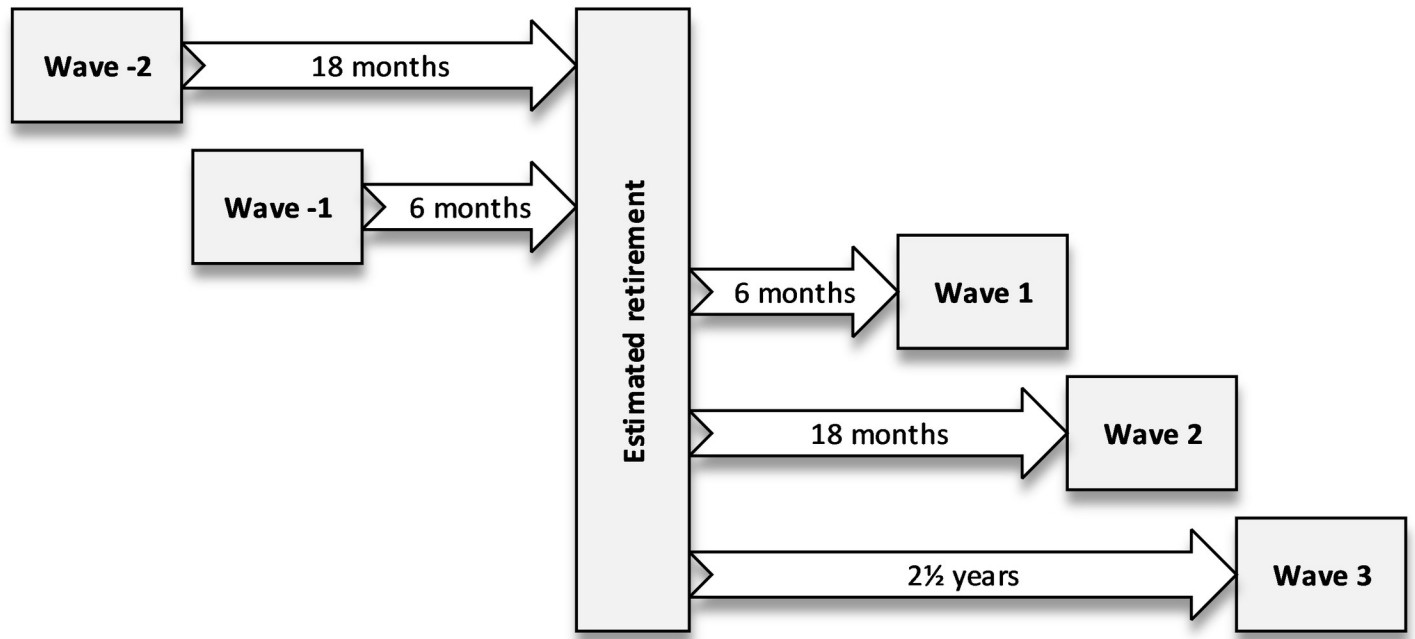

**Fig 1. Study waves and follow-up time.**

## Measurement of physical activity and body mass index

Physical activity was self-reported at each study wave. The participants were asked to estimate how often (not at all; less than 30 minutes; one hour; two to three hours; or over four hours) they were weekly engaged in activities comparable to 1) walking; 2) brisk walking; 3) jogging; or 4) running. The level of physical activity was then converted into metabolic equivalent of task (MET). The MET describes the amount of consumed energy comparing to resting. One MET unit of 3.5 ml/kg/minute corresponds to oxygen consumption while sitting at rest. Weekly physical activity was expressed as MET-h/week. For the interpretation of the results following categorization was used: low (<14 MET-h/week), moderate (14 to <30 MET-h/week) or high (≥30 MET-h/week) physical activity levels [11,12]. This categorization was chosen since physical activity higher than 14 MET-h/week has been reported to be associated with cardiovascular disease [13] and the activity level of 30 MET-h/week has been shown to be needed for weight management [14]. 14 MET-h/week is approximately the equivalent of 140 minutes of brisk walking weekly.

The respondents reported their body weight and height at each study wave. When interpreting the results, BMI was categorized as normal weight (18.5 to 24.9 kg/m$^2$); overweight (25 to 29.9 kg/m$^2$) or obesity (≥ 30 kg/m$^2$). Obesity was further categorized as type I (BMI 30 to 34.9 kg/m$^2$), type II (BMI 35 to 39.9 kg/m$^2$) or type III (BMI >40 kg/m$^2$) obesity. Underweight participants with BMI <18.5 kg/m$^2$ (n = 13) were excluded from the analysis.

## Demographics

Age was defined in full years and marital status was dichotomized as married or co-habiting vs. single at the last available wave before retirement (wave -1). Occupational titles were obtained from the register of pension provider and they were coded according to the International Standard Classification of Occupations (ISCO). According to the last known occupation prior to retirement, occupational status was dichotomized as professionals (ISCO major groups 1–4) vs. service and manual workers (ISCO major groups 5–9).

## Statistical analysis

The demographic estimates were reported as means and standard deviations or as absolute numbers and percentage, when appropriate. Group-based multi-trajectory analysis (GBTA) was used to investigate the developmental trajectories (a course of outcome over time) of physical activity and BMI. The group-based multi-trajectory analysis is a form of finite mixture modelling for analysing longitudinal repeated measures data [15]. While conventional statistics show a trajectory of average change of outcome over time, group-based trajectory modelling is able to distinguish and describe subpopulations (clusters) existing within a studied population. The trajectories of such subpopulations may differ substantially from each other and from the average trajectory observed in the entire population. The censored (known also as 'regular') normal model of group-based multi-trajectory analysis was used. The goodness of model fit was judged by running the procedure several times with a number of subpopulations starting from one up to five. The Bayesian Information Criterion (BIC), Akaike information criterion (AIC) and average posterior probability (APP) were used as criteria to confirm the goodness of fit. Linear, quadratic and cubic regression models were tested and cubic model was retained for using in the analysis. The cut-off for the smallest group was set at ≥5% of the entire cohort.

Multinominal regression analysis was used to describe the associations of demographic factors and probability of being classified into a particular cluster. The results were presented as odds ratios (OR) and their 95% confidence intervals (95% CIs).

The analyses were performed using Stata/IC Statistical Software: Release 16. College Station (StataCorp LP, TX, USA). The additional Stata module 'traj' was required to conduct group-based trajectory analysis. The module is freely available for both SAS® and Stata software [16].

## Results

Chracteristics of the study population are shown in Table 1. Of the respondents, 83% were women. The mean age before retirement, at the wave -1, was 63.3 (SD 1.4) years, 70% were married or co-habiting. Of the respondents, 64% were professionals and 36% were service or manual workers.

### Trajectory groups

A four-trajectory model was chosen as the smallest group in the more detailed five-trajectory model fell below a pre-agreed cut-off of 5% (Table 2). The average estimates of physical activity and body mass index at different study waves are shown by trajectory groups in S1 Table.

Four concurrent trajectory groups of physical activity and BMI were described as follows (Fig 2):

### Group 1: Individuals with normal weight and high level of physical activity (32%)

The BMI estimates remained stable around 22 kg/m$^2$ throughout the follow-up. Initially high physical activity of 31 MET-h/week slightly decreased prior to the retirement, subsequently increased by 2.7 units, and eventually stabilised at around 33 MET-h/week, being mildly higher than before retirement.

### Group 2: Individuals with overweight and moderately high level of physical activity (39%)

The BMI estimates remained stable around 26 kg/m$^2$ throughout the follow-up. Initially moderate physical activity of 25 MET-h/week slightly decreased prior to retirement, subsequently

**Table 1. Characteristics of the study population at pre-retirement (wave -1).**

| | *N* | *%* |
|---|---|---|
| **Gender** | | |
| Women | 2,783 | 83 |
| Men | 568 | 17 |
| **Occupation** | | |
| Professional | 2,133 | 64 |
| Manual and service worker | 1,190 | 36 |
| **Marital status** | | |
| Married or co-habiting | 2340 | 72 |
| Single | 927 | 28 |
| **BMI (kg/m2)** | | |
| Normal weight ($\leq$25kg/m2) | 1,289 | 38.5 |
| Overweight (25 to 29.9kg/m2) | 1,338 | 39.9 |
| Obesity (>30kg/m2) | 724 | 21.6 |
| **Physical activity (MET-h/week)** | | |
| Low ($\leq$14 MET-h/week) | 1,263 | 38 |
| Moderate (14 to 30 MET-h/week) | 1,022 | 31 |
| High (>30 MET-h/week) | 1,066 | 32 |
| **Age *(Mean/95% CI)*** | 63.3 | *1.4* |

**Table 2. Goodness of fit of group-based trajectory analysis models.** The chosen model is shown in bold.

| Model | Shape of trajectory | Smallest group | | BIC[1] | AIC[2] | APP[3] |
|---|---|---|---|---|---|---|
| | | n | % | | | |
| 1-cluster | cubic | 3,351 | 100% | -89157.68 | -89117.18 | 1.0 |
| 2-cluster | cubic | 1,048 | 31% | -85086.16 | -85009.22 | 0.96 |
| 3-cluster | cubic | 382 | 11% | -82673.30 | -82559.91 | 0.95 |
| 4-cluster | linear | 205 | 6% | -81095.94 | -81010.90 | 0.94 |
| **4-cluster** | **cubic** | **207** | **6%** | **-81161.68** | **-81011.84** | **0.94** |
| 5-cluster | cubic | 66 | 2% | -79903.10 | -79716.82 | 0.94 |

[1] BIC = Bayesian Information Criterion, [2] AIC = Akaike information criterion, [3] APP = Smallest average posterior probability.

increased by 2 units during retirement transition, and then decreased again in post-retirement years back to the initial level of 25 MET-h/week.

### Group 3: Individuals with class I obesity and moderately high level of physical activity (23%)

The BMI estimates remained stable around 31 kg/m$^2$ throughout the follow-up. Initially moderate activity of 20 MET-h/week first increased during a retirement transition by 1.4 units and then slightly decreased returning to the initial level of 20 MET-h/week.

### Group 4: Individuals with class II obesity and low level of physical activity (6%)

The BMI estimates remained stable around 37 kg/m$^2$ throughout the follow-up. Initially low activity of 10 MET-h/week slightly increased during and after retirement transition by 2.3 units, and then decreased to the level of 11 MET-h/week, which was close to the level before retirement.

### Associations between demographic factors and trajectory groups

Compared to the group #1, there were less women in groups #2 (OR 0.48, 95% CI 0.38 to 0.60) and #3 (OR 0.63, 95% CI 0.49 to 0.83). For the living arrangements, the groups were relatively

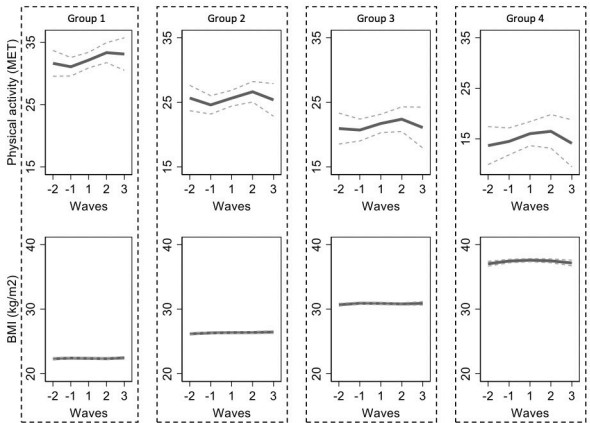

**Fig 2. Trajectories of BMI and physical activity in five study waves during retirement transition.** Each study wave is one year apart from each other.

**Table 3. Odds ratios of being classified to a particular group.** Group #1 used as a reference.

| Class 1 vs. Class 2 | N<br>Class1/2 | %<br>Class 1/2 | Odds Ratio | 95% CI | |
|---|---|---|---|---|---|
| **Women vs. men** | | | | | |
| Group #1 Stable normal weight and high physical activity | 943/125 | 88/12 | 1.0 | - | - |
| Group #2 Stable overweight and moderate physical activity | 1023/284 | 78/22 | 0.48 | 0.38 | 0.60 |
| Group #3 Stable class I obesity and moderate physical activity | 636/133 | 83/17 | 0.63 | 0.49 | 0.83 |
| Group #4 Stable class II obesity and low physical activity | 181/26 | 87/13 | 0.92 | 0.59 | 1.45 |
| **Co-habiting vs. single** | | | | | |
| Group #1 Stable normal weight and high physical activity | 746/298 | 71/29 | 1.0 | - | - |
| Group #2 Stable overweight and moderate physical activity | 946/325 | 74/26 | 1.13 | 0.95 | 1.35 |
| Group #3 Stable class I obesity and moderate physical activity | 531/220 | 70/30 | 0.96 | 0.79 | 1.18 |
| Group #4 Stable class II obesity and low physical activity | 117/84 | 58/42 | 0.56 | 0.41 | 0.76 |
| **Manual workers vs professionals** | | | | | |
| Group #1 Stable normal weight and high physical activity | 352/705 | 33/67 | 1.0 | - | - |
| Group #2 Stable overweight and moderate physical activity | 463/833 | 36/64 | 1.11 | 0.94 | 1.32 |
| Group #3 Stable class I obesity and moderate physical activity | 293/472 | 38/62 | 1.24 | 1.02 | 1.51 |
| Group #4 Stable class II obesity and low physical activity | 82/123 | 40/60 | 1.34 | 0.98 | 1.82 |

similar. The only exception was group #4, which was characterized by a lower proportion of co-habiting (OR 0.56, 95% CI 0.41 to 0.76) compared to the group #1. In terms of occupational group differences, the group #1 contained fewer manual workers than each of other three groups, but only group #3 showed statistically significantly difference compared to the group #1 (OR 1.24, 95% CI 1.02 to 1.51). The odds ratios of being classified to a certain group are presented in Table 3.

## Discussion

This prospective cohort study amongst 3,351 public sector employees investigated trajectories of concurrent changes in physical activity and BMI during transition to retirement. Four trajectory groups were identified, displaying stable BMI at normal, overweight, obesity I level and obesity II level, while the intensity of physical activity ranged from moderate or vigorous among normal weighted individuals to low physical activity among severely obese individuals. Consistently, when activity was low, BMI was high. Only physically very active participants were categorized into a normal weight trajectory group. Most of the participants were physically fairly active. Only the small group of obese participants did not meet the recommendations for physical activity [17,18]. Physical activity showed slight fluctuations around the retirement with small increase during retirement transition and return to initial level in post-retirement years.

In the current study population, 39% were normal weight, 40% overweight and 22% obese. The distribution of BMI estimates in this study was quite similar to the finding seen in some national studies among Finnish people, with little more normal weight and little less obese participants. Of the 60 to 69-year-old participants in the FinHealth -study from 2017, 30% were obese, 40% overweight and 30% had normal BMI [19]. On contrary, while 63% of the participants in this study population met the recommendations for physical activity, only 45% of the participants in a respective age group have met the same recommendations in the FinHealth -study [19]. This difference could be due to the fact, that most of the present participants were probably in good shape as they could keep on working until old-age retirement. The participants were also predominated by professional workers, who could have more assets to healthier lifestyle.

In line with previous research, the current study shows a little increase in leisure-time physical activity after retirement [7,8]. The initial pre-retirement activity level varied between the trajectory groups from low (10 MET-h/week) to high (31 MET-h/week), but the trajectory pattern during retirement transition was very similar in all groups. Similar small and only transient increase in physical activity has also been seen in previous works [7,8]. This passing increase could probably be explained by the increased amount of leisure time after the retirement, which could be used for physical activity [20]. People might be eager to try new hobbies, but then eventually return to their old habits. Also, with ageing, the probability of chronic medical conditions may increase [21], explaining a small decline in physical activity after the retirement transition.

The current study found BMI to be overall stable around retirement. This observation was in contrast to previous studies, which have showed a small increase in BMI from two to 10 years after the retirement, especially among manual workers [3,7]. This could be due to the sample consisting of mostly (64%) professional workers, among who the increase has not been established as clearly. BMI did not decrease even if physical activity increased. This could be due to dietary habits, which were not taken into account in the current study. Altered daily routine after retirement might change dietary habits, but the changes have been inconsistent. A Finnish study found decrease in vegetable consumption among women and increase in fruit consumption among men [22], while a study in continental Europe found decrease in fruit and vegetable consumption among men but no change in dietary habits among women [2]. Among manual workers, the absence of work-related physical activity might also lead to lower energy consumption and total activity even though leisure-time physical activity increased.

The current study also examined how demographic factors were associated with trajectory groups and some differences between the trajectory groups were found. The group with normal weight and physically highly active participants was used as a reference. The largest group consisted of overweight, but not obese, and physically active participants, who were more likely to be men than women. Slightly obese but fairly active participants, who were more likely to be men than women and service or manual workers than professionals, formed the 3rd largest group. The smallest group consisted of obese participants with low level of physical activity, who were more likely to be single.

Some previous studies have observed an association between overweight and marital status in men but not in women [23,24]. On contrary to previous research, in the current study being single predicted higher probability of being more obese and physically less active compared to co-habiting peers, whereas a previous study has reported that married persons are more likely to exercise less than singles (although, when ageing, this negative effect might turn into a positive one among men but not among women) [24].

According to consistent previous knowledge, normal weight is associated with higher educational level while increased leisure-time physical activity is associated with higher socioeconomic status [25,26]. A systematic review on the matter has stated that low socioeconomic status is consistently associated with lower activity around retirement [1]. Correspondingly, this study observed the association between higher (at least moderate) physical activity and higher occupational status.

The strengths of this study were a longitudinal repeated measures design and a large sample. Reflecting the common gender distribution in a Finnish public sector (high prevalence of female-dominated professions in healthcare and education), the cohort was predominated by women. Most of the participants were working until the old-age retirement. They have probably been more healthy and better functioning than employees who have retired earlier due to health issues. The study concerned leisure time and commuting activity, but changes in physical activity mainly reflect the changes in leisure time activity, since after retirement there is no

longer commuting activity. Previous research has stated that active commuters are more active after retirement than non-active commuters [9]. Regardless, for some people, total physical activity (including commuting) might have decreased even though the amount of leisure-time physical activity increased. If such is the case, then energy expenditure might not increase and BMI remains stable. The possibility of information bias should be taken into account due to the self-reported BMI and physical activity measures [27].

Further research is needed to address the reasons why the increase in physical activity, seen in this study, was only temporary and how to keep up the increased activity. Intervention studies are needed to see, which interventions would be significant to maintain a more active lifestyle after retirement. This knowledge may help to enhance interventions targeting improvement in physical activity behavior among retiring workers. This study used a group-based multi-trajectory analysis (GBTA). It is possible that other approaches to analyzing longitudinal data, like e.g., general linear model (GLM), structured equations model (SEM), latent growth or latent class analyses, may result in different findings. In further research, applying these models to similar datasets may provide additional valuable information on the topic.

## Conclusions

The results of this study observed a strong association between BMI and physical activity around retirement transition. Retirement seems to have more effect on physical activity than BMI, showing a temporary increase in physical activity at the time of retirement. Those who are single and manual workers were slightly more likely to be obese and physically less active. The findings suggest that there is potential for lifestyle changes and increasing activity around retirement, but more support is needed to maintain the increased activity level and more research is needed on how to make the changes constant. Current findings could be useful when planning interventions for people at the age around retirement.

## Supporting information

**S1 Table. Physical activity and body mass index at different waves by trajectory groups.**
(PDF)

## Author Contributions

**Conceptualization:** Roosa Lintuaho, Mikhail Saltychev, Jaana Pentti, Jussi Vahtera, Sari Stenholm.

**Data curation:** Mikhail Saltychev, Jaana Pentti, Jussi Vahtera, Sari Stenholm.

**Formal analysis:** Mikhail Saltychev, Jaana Pentti.

**Methodology:** Jaana Pentti.

**Supervision:** Mikhail Saltychev, Jussi Vahtera, Sari Stenholm.

**Validation:** Jussi Vahtera.

**Visualization:** Mikhail Saltychev.

**Writing – original draft:** Roosa Lintuaho.

**Writing – review & editing:** Mikhail Saltychev, Jussi Vahtera, Sari Stenholm.

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
