## [Decision Letter · Decision Letter 0]

26 Sep 2022

PONE-D-22-15861Multi-trajectory analysis of changes in physical activity and body mass index in relation to retirement: Finnish Retirement and Aging studyPLOS ONE

Dear Dr. Lintuaho,

Thank you for submitting your manuscript to PLOS ONE. After careful consideration, we feel that it has merit but does not fully meet PLOS ONE’s publication criteria as it currently stands. Therefore, we invite you to submit a revised version of the manuscript that addresses the points raised during the review process.

We look forward to receiving your revised manuscript.

Kind regards,

Giancarlo Condello, Ph.D.

Academic Editor

PLOS ONE

Journal Requirements:

Additional Editor Comment:

Dear Authors,

please follow the reviewers' comments and suggestion to improve your manuscript.

Best

Reviewers' comments:

Reviewer's Responses to Questions

**Comments to the Author**

1. Is the manuscript technically sound, and do the data support the conclusions?

Reviewer #1: Yes

Reviewer #2: Yes

2. Has the statistical analysis been performed appropriately and rigorously? 

Reviewer #1: Yes

Reviewer #2: Yes

3. Have the authors made all data underlying the findings in their manuscript fully available?

Reviewer #1: Yes

Reviewer #2: Yes

4. Is the manuscript presented in an intelligible fashion and written in standard English?

Reviewer #1: Yes

Reviewer #2: Yes

5. Review Comments to the Author

Reviewer #1: The manuscript entitled “Multi-trajectory analysis of changes in physical activity and body mass index in relation to retirement: Finnish Retirement and Aging study” examines a topic that is important in the context of ageing population and finding solutions to keep up physical functioning and healthy years during old age. The manuscript was very clearly written and the methods and results very well described. I have only minor comments / suggestions:

Abstract:

please state what is the actual follow-up time of the study.

Methods:

I suggest the authors add some kind of figure that connects the waves to chronological time and follow-up time.. now it is a bit unclear what the waves are and what is the follow-up time.

Results:

The legend of Figure 1 was missing, and the top line was not visible (there was a black box covering it). Thus, I was not sure what was the order of the groups in the figure. Make sure that the Figure is clear, and legends included in the final version.

Discussion:

How the trajectories compare to, for example, FinHealth 2017 or previous Finrisk studies for these age groups? This comparison may help to understand the generalizability of your results. The normal weight and high physical activity group is now the biggest group which makes me think, is this what we see at population level too, or is it just that the study sample consisted people who had higher education (who usually have healthier lifestyle and are leaner).

Lines 230-237: You have 5 years follow-up. How realistic did you consider that you would observe remarkable changes in BMI especially? Average weight gain on a “steady state” has been estimated to be 300g per year (https://pubmed.ncbi.nlm.nih.gov/21126985/)… thus this would mean 1.5kg over 5 years which is very difficult to detect with self-reported data. Of course, weight does not change in a consistent way and retirement could very well be a point where bigger changes could be expected. Further, could the stable BMI over the follow-up also stem from that most of the participants were professionals and not manual workers, among whom the increase in weight has been observed. You mention changes in dietary habits, that contribute to BMI. These were not assessed in your study, but previous studies have examined this, in case you want to discuss the role of diet more: https://pubmed.ncbi.nlm.nih.gov/31955724/

Line 257: I tried to go back to the Methods section, but I could not find information on the actual follow-up time. Based on the number of waves, I would say it was probably 5-6 years? If yes, then I would not call it as “long follow-up”. In epidemiology, a follow-up time of 20-30 years would be long especially when it is just questionnaire survey. I suggest you rephrase the sentence.

Perhaps it would be good to mention in the limitations that BMI and physical activity were self-reported.

Conclusion:

I was thinking that the main public health message (even though more research is needed) could be that retired people should be supported during the 5 years after retirement to keep up the improved lifestyle and help them to make the increased physical activity as a habit. Interventions could be arranged to study what the best support mechanism could be. Reduced prices to swimming halls, etc. do not seem to help in maintaining the motivation as these are already established in Finland. If the increment in physical activity would become permanent habit, it would probably have an impact on the functional capacity of these people as they get older, which again would have significant implications for elderly care and public economy. You are kind of hinting towards this now, but I think you could give stronger future perspective in your conclusion to highlight the importance of your observations.

Reviewer #2: This is a well written paper. The sample size is large, the sample is unique, the research question is noverl, and the results are new. These are advantages of the paper.

The problems include:

1- One size does not and should not fit all. Why subgroups are not analysed based on age (older old) or sex. If you rerun all the models only for women, do you replicate the same findings? What aout class?

2- The authors have used Group-based multi-trajectory analysis (GBTA) for their data analysis. They have not done other analyses such as repeated measure GLM, SEM, or latent growth, or latent class. Any of these approaches could result in different findings, and there is a need for some sensitivity analysis.

3- There is no mention of eduaction, income, poverty status, immigration (nativity), SES, ethnicity, or class. How these factors are not considered?

4- If BMI is not the main player in classes, then the main determinant of latent class is physical activity, not both of your variables. So, we need to see the class analysis when only physical activity is analysed. Do we see the same number of groups? If this is the case, BMI is a noise in this study. Please let us see through sensitivity analyses.

6. PLOS authors have the option to publish the peer review history of their article (what does this mean?). If published, this will include your full peer review and any attached files.

Reviewer #1: **Yes: **Noora Kanerva

Reviewer #2: No

---

## [Author Response · Author response to Decision Letter 0]

17 Oct 2022

Responses to comments from Reviewer #1:

Comment 1: Abstract: please state what is the actual follow-up time of the study.

Response 1: The follow-up time has been added to the abstract as follows:

“Repeated postal survey, including questions on physical activity and body weight and height, was conducted once a year up to five times before and after the retirement transition, the mean follow-up time being 3.6 years (SD 0.7).”

Comment 2: Methods: I suggest the authors add some kind of figure that connects the waves to chronological time and follow-up time.. now it is a bit unclear what the waves are and what is the follow-up time.

Response 2: Thank you for this suggestion. We have now added a figure (Fig 1) to clarify the study design. We have also modified the text as follows:

“There were two possible survey waves before the retirement, waves -2 (approximately 18 months prior to estimated retirement date) and -1 (approximately 6 months prior to retirement), and three possible waves after the estimated retirement date, waves 1 (approximately 6 months after retirement), 2 (approximately 18 months after retirement) and 3 (approximately 2.5 years after retirement) (Fig 1).”

Fig 1. Study waves and follow-up time

Comment 3: Results: The legend of Figure 1 was missing, and the top line was not visible (there was a black box covering it). Thus, I was not sure what was the order of the groups in the figure. Make sure that the Figure is clear, and legends included in the final version.

Response 3: We apologize for the unclear information. The Figure 1 and the legend (Fig 2: Trajectories of BMI and physical activity in five study waves during retirement transition. Each study wave is one year apart from each other) have been re-added.

Comment 4: Discussion: How the trajectories compare to, for example, FinHealth 2017 or previous Finrisk studies for these age groups? This comparison may help to understand the generalizability of your results. The normal weight and high physical activity group is now the biggest group which makes me think, is this what we see at population level too, or is it just that the study sample consisted people who had higher education (who usually have healthier lifestyle and are leaner).

Response 4: Thank you for pointing this out. The findings regarding BMI are quite similar to FinHealth 2017, but regarding physical activity there is difference. This has now been discussed in the study as follows: “ In the current study population, 39% were normal weight, 40% overweight and 22% obese. The distribution of BMI estimates in this study was quite similar to the finding seen in some national studies among Finnish people, with little more normal weight and little less obese participants. Of the 60 to 69-year-old participants in the FinHealth -study from 2017, 30% were obese, 40% overweight and 30% had normal BMI.[19] On contrary, while 63% of the participants in this study population met the recommendations for physical activity, only 45% of the participants in a respective age group have met the same recommendations in the FinHealth -study [19]. This difference could be due to the fact, that most of the present participants were probably in good shape as they could keep on working until old-age retirement. The participants were also predominated by professional workers, who could have more assets to healthier lifestyle.”

We have also added a table (Table 1) showing the characteristics of the study population at pre-retirement (wave -1).

Comment 5: Lines 230-237: You have 5 years follow-up. How realistic did you consider that you would observe remarkable changes in BMI especially? Average weight gain on a “steady state” has been estimated to be 300g per year (https://pubmed.ncbi.nlm.nih.gov/21126985/)… thus this would mean 1.5kg over 5 years which is very difficult to detect with self-reported data. Of course, weight does not change in a consistent way and retirement could very well be a point where bigger changes could be expected. Further, could the stable BMI over the follow-up also stem from that most of the participants were professionals and not manual workers, among whom the increase in weight has been observed. You mention changes in dietary habits, that contribute to BMI. These were not assessed in your study, but previous studies have examined this, in case you want to discuss the role of diet more: https://pubmed.ncbi.nlm.nih.gov/31955724/

Response 5: Thank you for this comment. 

Some previous studies have shown that retirement is associated with gaining weight, mostly among manual workers. The limit of this study is that BMI and physical activity were self-reported, and the possibility of information bias should be taken into account, as mentioned in the final rows of discussion; “The possibility of information bias should be taken into account due to the self-reported BMI and physical activity measures.” 

As for the stable BMI being due to the sample consisting mostly of professional workers, this has now been discussed in the Discussion as follows: “This could be due to the sample consisting of mostly (64%) professional workers, among who the increase has not been established as clearly.”

We thank you for presenting us with the study of dietary habits and it has now been mentioned in the discussion as follows: “Altered daily routine after retirement might change dietary habits, but the changes have been inconsistent. A Finnish study found decrease in vegetable consumption among women and increase in fruit consumption among men[22], while a study in continental Europe found decrease in fruit and vegetable consumption among men but no change in dietary habits among women[2].”

Comment 6: Line 257: I tried to go back to the Methods section, but I could not find information on the actual follow-up time. Based on the number of waves, I would say it was probably 5-6 years? If yes, then I would not call it as “long follow-up”. In epidemiology, a follow-up time of 20-30 years would be long especially when it is just questionnaire survey. I suggest you rephrase the sentence.

Response 6: Thank you for this comment. The sentence has been rephrased and the term “long follow-up” has been removed.

Comment 7: Perhaps it would be good to mention in the limitations that BMI and physical activity were self-reported.

Response 7: This is mentioned in discussion; “The possibility of information bias should be taken into account due to the self-reported BMI and physical activity measures.[26]”

Comment 8: Conclusion:

I was thinking that the main public health message (even though more research is needed) could be that retired people should be supported during the 5 years after retirement to keep up the improved lifestyle and help them to make the increased physical activity as a habit. Interventions could be arranged to study what the best support mechanism could be. Reduced prices to swimming halls, etc. do not seem to help in maintaining the motivation as these are already established in Finland. If the increment in physical activity would become permanent habit, it would probably have an impact on the functional capacity of these people as they get older, which again would have significant implications for elderly care and public economy. You are kind of hinting towards this now, but I think you could give stronger future perspective in your conclusion to highlight the importance of your observations.

Response 8: Thank you for this comment. We have made changes to the conclusions regarding these notes as follows: “Further research is needed to address the reasons why the increase in physical activity, seen in this study, was only temporary and how to keep up the increased activity. Intervention studies are needed to see, which interventions would be significant to maintain a more active lifestyle after retirement.” and “The findings suggest that there is potential for lifestyle changes and increasing activity around retirement, but more support is needed to maintain the increased activity level and more research is needed on how to make the changes constant.”

 

Responses to comments from reviewer #2:

Comment 1: One size does not and should not fit all. Why subgroups are not analysed based on age (older old) or sex. If you rerun all the models only for women, do you replicate the same findings? What about class?

Response 1: Thank you for pointing this out. The subgroups are not analyzed separately for age since there is not much difference in the age of the participants. The mean age 6 months prior to retirement (study wave -1) was 63.3 and the SD was 1.4 years. The age for old age retirement in Finland is usually 63 to 65 years, and due to eligibility criteria, the participants mostly retired on their planned retirement date, as they had to be still working during the first study wave and retired during the follow-up.

Sex, occupational class and marital status were used as demographic factors to see whether they affect the probability of being classified to a certain group, and thus they were not used to create the group-based multitrajectory analysis.

Comment 2: The authors have used Group-based multi-trajectory analysis (GBTA) for their data analysis. They have not done other analyses such as repeated measure GLM, SEM, or latent growth, or latent class. Any of these approaches could result in different findings, and there is a need for some sensitivity analysis.

Response 2: 

All the mentioned methods are well-suited for longitudinal data. However, our goal was not to conduct a methodological comparison of different statistical approaches. The GBTA was the method chosen for this study. We feel that such a comparison could be an interesting subject for further research, which recommendation has now been mentioned at the end of the discussion as follows: “This study used a group-based multi-trajectory analysis (GBTA). It is possible that other approaches to analyzing longitudinal data, like e.g., general linear model (GLM), structured equations model (SEM), latent growth or latent class analyses, may result in different findings. In further research, applying these models to similar datasets may provide additional valuable information on the topic”. 

Comment 3: There is no mention of eduaction, income, poverty status, immigration (nativity), SES, ethnicity, or class. How these factors are not considered?

Response 3: In this study, we used participants’ occupation as an indicator of their SES. Occupation was categorized as “professionals” and “service and manual workers” regarding the international classification of occupation (ISCO); classes 1 to 4 (e.g. doctors, teachers) were categorized as “professionals” and classes 5 to 9 (e.g. nurses, janitors etc.) The information on immigration or ethnicity was not available in the study, but in this cohort of Finnish aging workers responding to surveys in Finnish or Swedish (the official languages in Finland), immigrants and non-Caucasian ethnicity are very rare.

Comment 4: If BMI is not the main player in classes, then the main determinant of latent class is physical activity, not both of your variables. So, we need to see the class analysis when only physical activity is analysed. Do we see the same number of groups? If this is the case, BMI is a noise in this study. Please let us see through sensitivity analyses. 

Response 4: Thank you for this comment. In group-based multitrajectory analysis both of the variables, BMI and physical activity, are studied simultaneously and they are treated together. Rerunning the analysis separately for physical activity and BMI would probably lead to different results, because we aimed to study the concurrent, simultaneous changes of these two factors. For instance, if we ran the analysis separately we could find groups in which activity increases and BMI remains the same, but there would be no way of knowing, whether the same individuals experience these changes. In the current setting, we can see that, for instance in group #1 the individuals temporarily increase their high activity and keep their BMI stable.

---

## [Editor Report · Decision Letter 1]

16 Nov 2022

Multi-trajectory analysis of changes in physical activity and body mass index in relation to retirement: Finnish Retirement and Aging study

PONE-D-22-15861R1

Dear Dr. Lintuaho,

We’re pleased to inform you that your manuscript has been judged scientifically suitable for publication and will be formally accepted for publication once it meets all outstanding technical requirements.

Kind regards,

Giancarlo Condello, Ph.D.

Academic Editor

PLOS ONE
---

## [Editor Report · Acceptance letter]

21 Nov 2022

PONE-D-22-15861R1 

Multi-trajectory analysis of changes in physical activity and body mass index in relation to retirement: Finnish Retirement and Aging study 

Dear Dr. Lintuaho:

I'm pleased to inform you that your manuscript has been deemed suitable for publication in PLOS ONE. Congratulations! Your manuscript is now with our production department. 

Kind regards, 

on behalf of

Dr. Giancarlo Condello 

Academic Editor

PLOS ONE